# Targeting Class I-II-III PI3Ks in Cancer Therapy: Recent Advances in Tumor Biology and Preclinical Research

**DOI:** 10.3390/cancers15030784

**Published:** 2023-01-27

**Authors:** Benoît Thibault, Fernanda Ramos-Delgado, Julie Guillermet-Guibert

**Affiliations:** 1Centre de Recherches en Cancérologie de Toulouse, Université de Toulouse, Inserm, CNRS, 31037 Toulouse, France; 2LABEX TouCAN, 31037 Toulouse, France; 3Oncopole de Toulouse, CRCT UMR1037 INSERM-Université Toulouse 3, 2 Avenue Hubert Curien, CS 53717, CEDEX 1, 31037 Toulouse, France

**Keywords:** PI3K, cell signaling, cancer, combi/hybrid molecules, combination therapeutics

## Abstract

**Simple Summary:**

The PI3K/AKT pathway is one of the most important signaling nodes in cancer. While class I PI3K roles in cancer are well known, class II and III functions in physiology and physiopathology are poorly studied. Moreover, the interactions between pathways controlled by all three classes are not fully understood. The understanding of the mechanisms behind their cooperative functions could be key for efficient PI3K combination targeting in cancer therapy. This review will focus on the recent advances on the roles of the different classes of PI3K in cancer biology, their cross-regulations and their targeting in preclinical models.

**Abstract:**

Phosphatidylinositol-3-kinase (PI3K) enzymes, producing signaling phosphoinositides at plasma and intracellular membranes, are key in intracellular signaling and vesicular trafficking pathways. PI3K is a family of eight enzymes divided into three classes with various functions in physiology and largely deregulated in cancer. Here, we will review the recent evidence obtained during the last 5 years on the roles of PI3K class I, II and III isoforms in tumor biology and on the anti-tumoral action of PI3K inhibitors in preclinical cancer models. The dependency of tumors to PI3K isoforms is dictated by both genetics and context (e.g., the microenvironment). The understanding of class II/III isoforms in cancer development and progression remains scarce. Nonetheless, the limited available data are consistent and reveal that there is an interdependency between the pathways controlled by all PI3K class members in their role to promote cancer cell proliferation, survival, growth, migration and metabolism. It is unknown whether this feature contributes to partial treatment failure with isoform-selective PI3K inhibitors. Hence, a better understanding of class II/III functions to efficiently inhibit their positive and negative interactions with class I PI3Ks is needed. This research will provide the proof-of-concept to develop combination treatment strategies targeting several PI3K isoforms simultaneously.

## 1. Introduction

In humans, phosphatidylinositol-3-kinases (PI3Ks) comprise eight members that are divided into three classes. All PI3Ks catalyze a similar reaction, phosphorylation of phosphatidylinositols (PI) in position 3 of the inositol ring. However, they have different substrates, products and functions [1,2]. Class I PI3Ks are predominantly studied and play a critical role in cancer. The selective roles of those isoforms were exhaustively reviewed elsewhere [1,2,3]. Several inhibitors targeting class I PI3K are under development or have been clinically approved for cancer treatment. The current clinical application of PI3K inhibitors was recently reviewed by us and others [4,5]. On the contrary, class II and III PI3K physiopathological roles are still poorly understood. In the era of targeted therapies in cancer, it remains crucial to identify the specific PI3K isoform to target in each tumor to achieve an efficient clinical response (Figure 1). In this review, we will focus on the recent advances on the roles of the different classes of PI3K in preclinical models of cancer, their targeting and their cross-regulations. This knowledge is necessary and will inform our discussion on strategies to enhance the clinical efficacy of PI3K inhibitors, targeting one or several PI3K isoforms.

## 2. General Background on Clinical Use of Class I PI3K Isoform Inhibitors in Cancer and Rational of the Review

Class I PI3Ks comprise PI3Kα, PI3Kβ, PI3Kγ and PI3Kδ. Their respective catalytic subunits are encoded by PIK3CA, PIK3CB, PIK3CG or PIK3CD. PI3Kα and PI3Kβ are ubiquitously expressed while PI3Kγ and PI3Kδ are highly expressed in immune cells [1,2,3]. Low levels of PI3Kγ and PI3Kδ play selective roles in some cell types (e.g., vascular and lymphatic endothelial cells, cardiomyocytes). All class I PI3Ks phosphorylate phosphatidylinositol-4,5-bisphosphate (PIP2) into PIP3. The activity of class I PI3Ks can be reverted by the tumor suppressor PTEN, a phosphatase which hydrolyses PIP3 into PIP2. PIP3 next acts as a second messenger in the cell that activates the AKT/mTOR pathways [1].

The PI3K/AKT/mTOR pathway regulates multiple functions such as proliferation, migration, differentiation or metabolism [1]. It is now considered as one of the most mutated pathways in cancer, justifying the development of inhibitors of its upstream members, such as PI3Ks [6].

The first clinical trials using inhibitors targeting all PI3K isoforms (pan-PI3K inhibitors) in monotherapy showed disappointing results due to toxicities and resistance to treatment [4]. Indeed, the use of the pan-PI3K inhibitors LY-294002, PX-866, BKM-120, GDC-0941 and NVP-BEZ235 triggers an AKT reactivation after transient inhibition in colorectal tumor cells [7] or in pancreatic cancer cell lines [8]. In non-small cell lung cancer cell lines, the pan-PI3K inhibitors BKM-120, LY294002 and NVP-BEZ235 are responsible for the increase of MET tyrosine kinase expression and activation. This leads to the activation of signal transducer and activator of transcription 3 (STAT3) and treatment failure [9]. Moreover, combining pan-PI3K inhibitors with classical chemotherapies or other targeted therapies heightens the risk of multiple adverse effects [4].

To decrease feedback responses and toxicity events, isoform-specific PI3K inhibitors have been and are being developed. They may be tolerated at doses resulting in more complete target inhibition with less resistance and fewer adverse effects. This therapeutic strategy was supported by the fact that, even though all class I PI3Ks catalyze the same reaction, they show distinct relevance in each organ and in each tumor type [2]. The first two inhibitors that were approved for clinical use were PI3Kδ and PI3Kα-specific/selective inhibitors [10,11]. Later, pharmacological research allowed the development of dual- or triple-selective inhibitors; the latter being called isoform-sparing inhibitors. The rationale behind this development is based on findings demonstrating that while each PI3K isoform has selective roles, they also have synergistic actions in a tumor-intrinsic or extrinsic manner and/or can be re-activated as an adaptive response to PI3K isoform-specific inhibition. Hence, it becomes clear that PI3K targeting strategies should aim to precisely target PI3K isoforms in combination depending on their relative importance in each tumor. To reach this aim, recent efforts have been put forward to understand which mutational or expression profiles could lead the choice of using PI3Kα and PI3Kβ inhibitors.

The aim of the review is to provide state-of-the art knowledge on those interconnections between isoforms focusing mainly on preclinical models.

## 3. Mutation Profiles of Cancer Cells and Sensitivity to PI3Kα or PI3Kβ Inhibitors, Related Issues

PI3Kα and PI3Kβ are ubiquitously expressed but have different functions and relative importance in organs and cancers [3]. The importance of genetic alterations to trigger PI3Kα or PI3Kβ dependency is the most described and understood mechanism of selective PI3K activation. However, they do not predict nor fully explain the sensitivity to PI3K inhibitors (e.g., for PI3Kα inhibitors: [12,13]), suggesting that this simple model might not be enough to explain the sensitivity to isoform-selective inhibitors.

### 3.1. Oncogenic Mutation of PIK3CA and Sensitivity to PI3Kα Inhibitors

The PI3Kα-specific inhibitor alpelisib (PIQRAY or BYL-719) recently obtained approval for *PIK3CA*-mutated hormone receptor-positive advanced breast cancer in combination with fulvestrant, a hormonotherapy [11]. A new generation of PI3Kα-specific inhibitors has been developed with GDC-0077 (inavolisib) [14]. This drug shows strong anti-tumor activity in breast cancer by inducing the degradation of the specific mutant-p110α (a catalytic subunit of PI3Kα) in an HER2-dependent manner [14]. This result, based on years of preclinical research [15], demonstrates that PI3K inhibitors are efficient in the clinic when the feedback mechanism upon their inhibition is targeted and is a main driver of oncogenesis. The genetic mutation of the PI3Kα encoding gene is not enough to drive tumor progression; a context-dependent activation of PI3Kα is needed.

Preclinical models and early phase clinical trials demonstrate that PI3Kα-specific inhibitors exhibit anti-tumor efficacy in cancers with frequent *PIK3CA* oncogenic mutations, such as lung and colon cancers, but also in hematological cancers such as juvenile myelomonocytic leukemia, acute myeloid leukemia (AML) or chronic lymphocytic leukemia (CLL) [4]. Likewise, PI3Kα-specific inhibition induces cell death in *PI3KCA* mutated endometrial cell lines; the clinical relevance of this finding is still unclear [16].

### 3.2. Context-Dependent Sensitivity to PI3Kα Inhibitors

Preclinical investigations showed that PI3Kα inhibitors have higher target inhibition in cell lines and xenografts with *PIK3CA* alterations; these findings were confirmed in the clinic in metastatic breast cancer (MBC) patients undergoing hormonotherapy [11,15]. Our recent study suggests that in contrast to MBC, the PI3K pathway could drive pancreatic cancer metastatic progression despite the absence of *PIK3CA* oncogenic mutations. Patients with a PI3Kα activation gene signature are enriched in the most aggressive subtype of pancreatic ductal adenocarcinoma (PDAC) patients. Pharmacological and genetic PI3Kα inactivation demonstrate that PI3Kα is crucial for the development, progression and metastasis of pancreatic cancer [17]. This cancer presents a low frequency of *PIK3CA* mutations (5%) but a high percentage of *KRAS* oncogenic alterations (90%), responsible for the global activation of the PI3K/AKT pathway [18].

Interestingly, the detection of a *PIK3CA* mutation in circulating tumor DNA in MBC patients treated with alpelisib was linked to significantly improved progression-free survival under treatment, suggesting a context-dependent activation of PI3K in metastatic processes [19].

Additionally, other mutations/genetic alterations might need to be targeted to improve the efficiency of PI3Kα-specific inhibitors. Indeed, in most cases, PI3Kα inhibitors are more efficient in combination strategies. In prostate cancer, *PIK3CA* oncogenic mutations were found to cooperate with PTEN loss to accelerate tumor growth and facilitate castration resistance [20]. In pancreatic cancer, we found that co-treatment with PI3Kγ inhibitors prevents the feedback re-activation of AKT after PI3Kα inhibition [8]. In rhabdomyosarcoma cells, the combined inhibition of PI3Kα and δ, using alpelisib and idelalisib, respectively, synergistically inhibited cell survival, highlighting the potential of targeting several class I PI3K isoforms [21]. In the case of multi-isoform inhibitors, genomic analyses of the PI3Kβ-sparing taselisib-treated patients identified mutations potentially associated with an upfront resistance to PI3K inhibition (*TP53* and *PTEN*) and post-progression through the reactivation of the PI3K pathway (*PTEN*, *STK11* and *PIK3R1*) [12], possibly leading to an activation of PI3Kβ in this context. In Philadelphia chromosome-positive CLL, the PI3Kα-specific inhibitor alpelisib exhibited a synergy with Bcr-Abl-targeted drugs [22]. Alpelisib also showed synergistic anti-tumor activity with paclitaxel in gastric cancer with higher effects in *PIK3CA*-mutant cells [23]. A phase I clinical trial showed an indication of a synergy between the PI3Kα-specific inhibitor alpelisib and the PARP inhibitor olaparib in epithelial ovarian cancer [24].

PI3Kα inhibition not only affects tumor cells but also modulates the tumor microenvironment. The PI3Kα-specific inhibitor CYH33 enhances CD4^+^ and CD8^+^ T cell infiltration, inhibits the proliferation of M2 pro-tumoral macrophages and activates the differentiation of macrophages into an M1 phenotype. Likewise, CYH33 accelerates fatty acid metabolism in the tumor microenvironment which activates CD8^+^ T cell activity. The combined inhibition of fatty acid synthase (FASN) and PI3Kα triggers a synergistic anti-tumor effect in 4T1 breast cancer cells, further reinforcing the need to use isoform-specific PI3K inhibitors in combination with other targeted therapies in cancer [25].

### 3.3. Loss of PTEN and Limited Sensitivity to PI3Kβ Inhibitors when Used in Monotherapy

PTEN-deficient tumors rely mainly on PI3Kβ rather than PI3Kα [3], yet the underlying mechanisms of such selectivity are still unclear. The inhibition of PI3Kβ by AZD8186 in PTEN-null tumors modifies metabolic pathways by inhibiting the expression of enzymes from the cholesterol biosynthesis pathway [26]. These data suggest that the codependency of pro-tumoral actions due to PIK3β activation and PTEN loss could be attributed to their common activities in metabolic pathways.

Clinical research related to the use of PI3Kβ inhibitors is more advanced in prostate or breast cancers; nonetheless, PI3Kβ-specific inhibitors are promising therapies for other PTEN-deficient cancers such as melanoma, glioblastoma and bladder. Similar to PI3Kα targeting, PI3Kβ targeting seems to be more clinically efficient when used in combination treatments. Indeed, a phase I/II clinical trial evaluating the PI3Kβ inhibitor AZD8186 as a monotherapy or in combination with abiraterone acetate, an androgen inhibitor, showed preliminary data of anti-tumor activity in metastatic castrate-resistance prostate cancer [27]. In preclinical models of PTEN-deficient tumors, PI3Kβ inhibitors (AZD8186, AZD6482) yielded a higher efficacy when administered in combination with other therapeutic agents such as the mTOR inhibitor (prostate and other cancers [28]), PI3Kα inhibitor and fulvestrant (ER-positive breast cancers, [29]), paclitaxel and anti-PD1 treatment (triple negative breast cancer, [30]), epidermal growth factor receptor (EGFR) inhibitors (triple negative breast cancer, [31]), inhibitor of mixed-lineage protein kinase 3 (MLK3) (glioblastoma, [32,33]), MEK inhibitor selumetinib (malignant mesotheliomas, [34]) or antibodies directed against OX40, a T-cell costimulatory receptor melanoma [35].

### 3.4. Context-Dependent Sensitivity to PI3Kβ Inhibitors

PTEN genetic alterations are not solely responsible for the dependency on PI3Kβ inhibitors. Other teams have demonstrated the relevance of PI3Kβ in breast cancer metastasis, independently from its PTEN status [3,36]. Genetic alterations in *PIK3CB*, the gene encoding for the PI3Kβ catalytic subunit, could lead to a dependency on PI3Kβ signaling. A phase I clinical trial assessing the highly selective PI3Kβ-specific inhibitor GSK2636771 in PTEN-deficient advanced solid tumors showed a manageable safety profile with evidence of clinical activity in tumors harboring activating mutations in *PIK3CB* [37].

Though genetic mutations and epigenetic alterations of class I PI3Ks can result in their overactivation, the tumor context might also be permissive for its activation in the absence of oncogenic PI3K mutants. Recently, we highlighted the role of PI3K in mechanotransduction, a process in which mechanical forces (tensile, shear and compression) are transduced in biochemical signals [38]. Knowledge on PI3K isoform selectivity in mechanotransduction will allow the development of clinical applications using small molecule inhibitors in selective mechanical contexts. For example, the increase of YAP/TAZ gene signature transcriptional activity (a classical marker of tensile stress) is predictive of PI3Kβ isoform-selective KIN-193 efficacy in cancer [39]. Similarly, E-cadherin is a critical component of normal epithelial cell/cell adhesion and is known to inhibit YAP/TAZ, while the loss of E-cadherin expression in cancer increases sensitivity to PI3Kβ inhibition [40]. PI3Kα-driven YAP/TAZ activation, possibly through EGF-EGFR-PI3K or FAK–Src–PI3K activation sequences [41,42], is insufficient to promote tumor formation, whereas PI3Kβ-driven YAP/TAZ activation, likely via pertussis toxin-sensitive GPCRs-PI3K activation [43], allows tumor formation [44]. Overall, these findings could pave a way to novel leads of the predictive markers of PI3Kβ inhibitor efficiency.

In sum, PI3Kα and PI3Kβ have different roles in organs or in cell types (e.g., in endothelial cells [45] and in germinal cells [46]), but also in tumors with the same genetic alteration (*PTEN* heterozygous loss [47]). Their roles in cancer depending on the mutational context are now better understood (Figure 2). *PIK3CA* mutations or alterations in the tyrosine kinase receptor EGFR or KRAS signaling pathway would direct the treatment strategy towards drugs targeting PI3Kα. Meanwhile, the loss of PTEN would lead to approaches using PI3Kβ-specific inhibitors. Similarly, PTEN loss in breast cancer is responsible for the resistance to the PI3Kβ-sparing-inhibitor taselisib and CDK4/6 inhibitors [48]. At the moment, these proposed rules are insufficient to determine the appropriate administration of PI3K isoform-selective inhibitors. In a recent study, we find that even PTEN-depleted PDAC cell lines require a minimal PI3Kα activity to migrate [17]. The tumorigenic action of the knock-out of *PTEN* in the thyroid is protected by PI3Kα but not by PI3Kβ genetic inactivation [47]. The context-specific activation of PI3Kα or PI3Kβ in cancer requires further studies and characterization (Figure 2).

While the choice of using PI3Kα or PI3Kβ inhibitors was lately made depending on the mutational context, PI3Kγ and PI3Kδ remain the target of choice in hematologic cancers due to their high expression in these cell types.

## 4. PI3Kδ and PI3Kγ, Their Key Tumor Intrinsic and Extrinsic Roles in Hematologic Cancers

The first PI3K inhibitor to be clinically approved was idelalisib, a PI3Kδ-specific inhibitor that showed anti-tumor activity alone in relapsed indolent lymphoma, or in combination with rituximab (CD20 inhibitor) in chronic lymphocytic leukemia [5,10,52]. Idelalisib, which targets PI3Kδ in the BCR pathway, only generates a partial response in CLL patients, though the concomitant inactivation of PI3Kβ reduces further downstream activation in patient-derived CLL cells [53]. Recent data in murine models of CLL show that PI3Kδ-specific inhibition with idelalisib induces treatment resistance which could be overcome by targeting the insulin-like growth factor 1 receptor (IGF1R) [54]. Similarly, the combination of idelalisib and arsenic trioxide showed a synergy in human acute promyelocytic leukemia NB4 cells when compared to each treatment alone, accentuating the interest to use this PI3Kδ-specific inhibitor in combination in hematological diseases [55].

Other PI3Kδ-specific inhibitors, such as umbralisib, have thus been developed for hematological cancers and are currently being tested in clinical trials. In a phase I clinical trial, umbralisib showed promising signs of clinical efficacy in relapsed or refractory chronic lymphocytic leukemia, small lymphocytic lymphoma, B-cell and T–cell non-Hodgkin’s lymphoma, Hodgkin’s lymphoma and relapsed or refractory indolent lymphoma [56,57]. This inhibitor (also named TGR-1202) synergizes with carfilzomib, a proteasome inhibitor, by reducing c-Myc translation [58]. Umbralisib approval has been recently withdrawn due to high toxicities outweighing the benefits of the treatment, raising the question of its appropriate dosage in patients.

Inhibitors of multiple PI3K isoforms have been developed to mitigate toxicity issues and improve efficacy. In a phase II clinical trial, copanlisib, a PI3K inhibitor targeting preferentially α and δ isoforms, showed clinical efficacy in patients with indolent or aggressive malignant lymphoma [59].

Besides directly targeting lymphoma cells, PI3Kδ/γ inhibitors could act through immunomodulatory actions. A phase I clinical trial showed the promising clinical activity of duvelisib, a PI3Kδ/γ inhibitor in T-cell lymphoma, by reprogramming macrophages (cells enriched in PI3Kγ) and inducing tumor cell-autonomous death [59,60]. In CAR T cell therapy, the ex vivo treatment of T cells with a PI3Kγ/δ inhibitor reduced anti-tumor efficacy, while the single inhibition of each isoform generated cells with higher anti-tumoral activity [61]. These data clearly emphasize the relevance of targeting PI3Kγ or PI3Kδ individually in this context.

Even though PI3Kγ and PI3Kδ roles were first described in immune cells, their overexpression in solid tumors and their potential role in tumor stroma make them promising isoforms to target in all cancers.

## 5. PI3Kδ and PI3Kγ, Unexpected Roles in Solid Tumors

PI3Kδ is mainly expressed in white blood cells in physiological conditions; nonetheless, its expression has been found in other solid tumors such as liver or breast cancer. In hepatocellular carcinoma, the genetic or pharmacological inhibition of PI3Kδ by idelalisib reduces tumor progression and induces apoptosis via Bim [62,63]. In breast cancer, the pharmacological targeting of PI3Kδ inhibits tumor cell migration and prevents tumor progression by acting on cancer cells and macrophages [64].

A recent clinical trial demonstrated a novel clinical application of the PI3Kδ inhibitor AMG319 in solid cancer (head and neck), through its immunoregulation role. This clinical trial also showed that a modified treatment regimen with the intermittent dosing of a PI3Kδ inhibitor in mouse models led to a significant decrease in tumor growth without inducing autoimmune colitis through the depletion of regulatory T cells in colonic tissue [65], suggesting that alternative dosing regimens might limit toxicity.

Similarly, PI3Kγ might harbor tumor-intrinsic and tumor-extrinsic roles in cancer. PI3Kγ expression was described as specific to the immune cell lineage. Despite that, several laboratories including ours suggest that the expression of PI3Kγ in tumor cells, albeit not strong, plays a critical role in KRAS^G12D,G12V^ pancreatic tumor cells [8,51], but also in KRAS^G12R^ pancreatic cells (a rarer form of mutant KRAS found in PDAC) [66]. In this latter genetic context, PI3Kγ is not directly activated by KRAS but sustains macropinocytosis, a cellular process that promotes the non-selective uptake of extra-cellular material such as solute molecules, nutrients and antigens in harsh conditions. In pancreatic cancer, the long-term inhibition of PI3Kγ with specific inhibitors induces compensation by other class I PI3K isoforms. In vivo, the targeting of PI3Kα and PI3Kγ using the specific inhibitors BYL-719 and IPI-549, respectively, showed synergistic anti-tumor effects compared to each inhibitor alone. These results confirm the importance of PI3Kγ in pancreatic cancers [51] and highlight the interactions existing between several class I PI3K isoforms [8]. Furthermore, PI3Kγ has recently been identified as an indirect or direct target for retinoblastoma. Indeed, the direct inhibition of its expression using siRNA or indirect inhibition via CANT1 long non-coding RNAs reduced retinoblastoma tumor growth [67].

Targeting PI3Kγ in solid tumors due to its roles in the immune microenvironment has shown promising results. In pancreatic cancer, the combined inhibition of PI3Kγ and colony-stimulating factor-1 receptor (CSF-1R) modified the macrophage balance by decreasing M2 pro-tumor macrophages and increasing M1 anti-tumor macrophages, thus reducing the tumor volume in mice [68]. Such a strategy currently tested in a phase I clinical trial in advanced solid tumors using the PI3Kγ-specific inhibitor IPI-549 in monotherapy or in combination with nivolumab, a programmed cell death protein 1 (PD1) inhibitor, showed promising clinical results with favorable tolerability [69]. In poorly immunogenic head and neck squamous cell carcinoma mice models, the genetic inhibition of *Pik3cg*, the gene encoding for the PI3Kγ catalytic subunit, did not alter tumor growth nor lymph node metastasis. However, those mice showed an increased CD8^+^ T-cell tumor infiltration associated with increased IFN-γ, IL-17 and PD-1 expression, suggesting that PI3Kγ inhibition could synergize with immunotherapies targeting PD-L1 in this cancer [70].

Lastly, PI3Kγ inhibition protects from cardiotoxicity induced by doxorubicin by inducing mitophagy triggered by this anthracycline. Moreover, the inhibition of PI3Kγ with the specific inhibitor AS605240 synergizes with the doxorubicin anti-tumor effect by reducing the recruitment of pro-tumoral macrophages in mice models of breast cancer [71].

PI3Kδ and PI3Kγ are now considered the targets of choice in solid tumors, alone or in combination with existing treatments because of their capacity to modulate the immune tumor microenvironment and direct tumor-cell intrinsic actions (Figure 1).

The roles of class I PI3Ks in cancer have been extensively studied and are now better understood, while class II and III PI3Ks remain unexplored.

## 6. Class II and III PI3Ks, Novel Possibilities to Target Cancer Progression

### 6.1. Class II PI3Ks and Organismal Functions

Class II PI3Ks consist of three isoforms, PI3K-C2α, PI3K-C2β and PI3K-C2γ, which synthetize in vitro and in cellulo two different products, PIP and PIP2, depending on the substrates. Class II PI3Ks have been largely neglected and their specific functions (Table 1), especially in cancer, are still poorly understood [72] (Figure 1).

Class II PI3Ks are monomeric enzymes consisting of a PI3K catalytic core extended by an N-terminus, a C-terminal phox homology (PX) domain and their characteristic C2 domain at the carboxyl terminus. Class II PI3K-C2α and PI3K-C2β are ubiquitously expressed, while PI3K-C2γ expression is restricted to the liver, breast, prostate, salivary glands and exocrine pancreas [73,74]. The lipid products of class II PI3Ks include PI_(3)_P and PI_(3,4)_P2 which are found in intracellular vesicles [75]. Class II PI3Ks mainly regulate intracellular dynamics, membrane trafficking, instead of acting as signal transducers like class I PI3Ks [1]. Knowledge on signaling through agonist dependent pools generated by class II PI3Ks is currently scarce and has only been described in specific settings such as insulin signaling: the role of PI3K-C2α in insulin signaling [76,77] and the role of PI3K-C2γ as a Rab5 effector [74]. Limited data on the modus operandi of class II PI3K has delayed the development of specific inhibitors; consequently, most of the information on their physiological functions has been obtained through genetic studies involving the knockdown of gene expression [78]. The use of genetically engineered mouse models (GEMMs) for class II PI3Ks has been crucial to determine their physiological roles. While PI3KC2α is essential in mice, human patients with inherited homozygous null mutations in the *PIK3C2A* gene display congenital syndromic features, including kidney failure and cataracts, due to early senescence of cells in the associated tissues [79,80]. This suggests that in humans, *PIK3CA* loss could be compensated with PI3KC2β encoded by *PIK3C2B*. Current published GEMMs are described in the following table (Table 1). Present GEMMs and genetic knockdown models have contributed to characterizing specific functions for class II PIKs and allowed the identification of further scaffolding properties of certain class II PI3Ks.

**Table 1 cancers-15-00784-t001:** Genetically engineered mouse models of class II PI3Ks (only systemic KO and KI mouse models were included in the table).

PI3K	Genotype	Phenotype	Reference
**PIK3C2A** **PI3K-C2α**	PI3K-C2α^−/−^ KO	Embryonic lethal E10.5 Defective vascularity	[81]
**PIK3C2A** **PI3K-C2α**	PI3K-C2α^−/−^ KO	Embryonic lethal E10.5 Delayed development from E8 Defective vascularity Defective sonic hedgehog signaling	[82]
**PIK3C2A** **PI3K-C2α**	PI3K-C2α^D1268A/wt^ KI	Only heterozygous are viable Metabolic defects in males: early onset leptin resistance and age-dependent obesity	[83]
**PIK3C2B** **PI3K-C2β**	PI3K-C2β^−/−^ KO	Viable No overt phenotype	[84]
**PIK3C2B** **PI3K-C2β**	PI3K-C2β^D1212A/D1212A^ KI	Viable and fertile Enhanced insulin sensitivity and glucose tolerance Resistance to liver steatosis under high-fat diet	[85]
**PIK3C2G** **PI3K-C2γ**	PI3K-C2γ^−/−^ KO	Viable and fertile Age-dependent insulin resistance Defective insulin response Increased obesity and fatty liver	[74]

### 6.2. Class II PI3Ks and Their Role in Cancer

PI3K-C2α is mainly described as regulating cancer cell death and mitosis [86,87], especially by controlling spindle stability [88]. PI3K-C2β is predominantly involved in cancer cell migration and invasion [89,90] but can also control proliferation via action on cyclin B1 expression [91]. The miRNA miR-362-5p can inhibit neuroblastoma cell proliferation and migration by targeting the 3′UTR of PI3K-C2β mRNA [92]. Current findings show that PI3K-C2β accelerates the disassembly of focal adhesion [93], a process that might be critical to drive metastatic dissemination. Likewise, data indicate that the inhibition of PI3K-C2β expression by siRNA delays mitosis in prostate cancer PC3 cells and potentiates the anti-clonogenic effect of docetaxel [94]. To date, PI3K-C2γ roles in cancer remain poorly described: the low expression of *PIK3C2G* is associated with an increased risk of recurrence and death in colorectal patients treated with oxaliplatin [95]. In pancreatic cancer, PI3K-C2γ expression is reduced in approximately 30% of pancreatic cancer cases. In mice models with pancreatic cancer, PI3K-C2γ loss is associated with an aggressive phenotype and an increased sensitivity to mTOR and glutaminase inhibitors [96]. In cancer settings, so far, the data from mouse models and human samples are concordant. In ovarian cancers, a nonsense mutation of *PIK3C2G* was identified but no functional analysis was performed [97]. Overall, these data suggest tissue-specific tumor suppressor functions of PI3K-C2γ that need be further confirmed. Notwithstanding, most of these works mostly describe in vitro experiments. More experimental work in integrated models or in more relevant cell models need to be performed to discern the specific roles of class II PI3K in different cancers.

As class II PI3Ks regulate key membrane-based cellular processes (through endocytosis, lysosomal activity or focal adhesion turnover), they might harbor pro- and anti-tumoral functions depending on the stage of the cancer process (the initiation vs. the progression stage). Playing a role in endothelial cell integrity [98], they might have key roles in tumoral vasculature. Their role in other cellular components of the tumor microenvironment also needs to be elucidated.

Ultimately, more in vivo data on the inactivation of those enzymes in a tissue- and time-specific manner and the development of class II-targeting drugs are needed. The latter is necessary to determine whether those compounds could strengthen the anti-tumoral PI3K-targeting arsenal.

### 6.3. Class III PI3K and Organismal Functions

Vacuole protein sorting 34 (Vps34) is the sole member of the class III PI3K and phosphorylates PI into PI-3-P. Vps34 is ubiquitously expressed and constitutes a heterodimer with Vps15 which has multiple physiological functions (Table 2). The Vps34/Vps15 complex is involved in complex I with Beclin-1 and ATG14, or complex II with Beclin-1 and UVRAG, playing a role in autophagy and endocytic sorting, respectively [1]. Vps34 participates in the regulation of autophagy, endocytosis and phagocytosis, which all converge at the lysosome degradation. During these processes, the activity of Vps34 is essential at the initial stages to produce the PI_(3)_P pools required for the biogenesis of autophagosomes, endosomes and phagosomes. Later on, it controls their maturation by recruiting effector proteins that promote their fusion with lysosomes [99,100]. Of note, in some physiological contexts such mechanical shear stress, autophagic pools of PI_(3)_P may also be synthesized by class II PI3Ks, including PI3KC2α [101]. Added to its functions in vesicle traffic, Vps34 has scaffolding properties. In several studies using Vps34-deficient mouse models, the animals presented a reduced expression of Vps34-binding partners, supporting the importance of the scaffolding function of Vps34 in maintaining the stability of the complexes [102]. Importantly, the phenotypes of Vps34 KO or of Vps15 KO do not fully match with the results obtained with the inactive mutant of Vps34 [1]. These findings accentuate the complexity when interpreting these phenotypes which might not be solely attributed to the catalytic activity of the enzyme, but also to its scaffolding properties. 

The following table (Table 2) recapitulates the available global KO and KI mouse models, as well as some examples of the main conditional mouse models of Vps34. It is important to mention that only the full deletion of Vps34 leads to important phenotypes in vivo, including severe cardiac effects [102]. For instance, the clinical use of Vps34 inhibitors requires a cautious approach to achieve a suitable therapeutic response.

**Table 2 cancers-15-00784-t002:** Genetically engineered mouse models of class III PI3Ks.

		Genotype	Phenotype	Reference
**Global**	**Models**	PIK3C3^−/−^ KO	Homozygous mice—embryonic lethal E7.5–E8.5 Heterozygous mice are viable and healthy, no overt phenotype	[103]
**Global**	**Models**	Vps34^D761A/+^ KI	Homozygous mice—Embryonic lethal E6.5–E8.5 Heterozygous mice are viable and fertile Heterozygous mice display enhanced insulin sensitivity and glucose tolerance	[104]
**Conditional** **Model**	**Deletion of exon 21** **(kinase domain-24 amino acids)**	PF4-Cre;Vps34^fl/fl^ platelets	Viable mice Abnormalities in platelets Impaired thrombus formation and granule secretion	[105]
**Conditional** **Model**	**Deletion of exon 21** **(kinase domain-24 amino acids)**	Pax8-Cre;Vps34^fl/fl^ proximal tubular cells (PTC)	Fanconi-like syndrome Vacuolation of PTCs	[106]
**Conditional model**	**Deletion of ATP-binding domain**	Advilin-Cre;PIK3C3^fl/fl^ neurons	Post-natal lethality after 2 weeks Neurodegeneration—vacuole formation in sensory neurons	[107]
**Conditional model**	**Deletion of ATP-binding domain**	CaMKII-Cre;PIK3C3^fl/fl^ pyramidal neurons	Loss of synapses Neurodegeneration Extensive gliosis	[108]
**Conditional model**	**Deletion of ATP-binding domain**	TgCKmm-Cre;PIK3C3^fl/fl^ cardiac and skeletal muscle	Post-natal lethality after 4 weeks Muscular dystrophy Cardiomyopathy	[109]
**Conditional model**	**Deletion of ATP-binding domain**	Pcp2-Cre;PIK3C3^fl/fl^ bipolar and Purkinje cells	Progressive degeneration of retinal bipolar cells and cerebellar Purkinje cells Reduced cerebella Progressive ataxia	[110]
**Conditional model**	**Deletion of ATP-binding domain**	Cone-Cre;PIK3C3^fl/fl^ retina cone cells	Progressive retinal degradation (onset 12 weeks) Loss of cone structure and degradation (1.5 months)	[111]
**Conditional model**	**Deletion of exon 4** **(N-terminus, any functional domain)**	Mck-Cre;PIK3C3^fl/fl^ heart	Post-natal lethality after 5 weeks Cardiomegaly	[102]
**Conditional model**	**Deletion of exon 4** **(N-terminus, any functional domain)**	Alb-Cre;PIK3C3^fl/fl^ liver	Post-natal lethality after 1 year Hepatomegaly Hepatic steatosis	[102]
**Conditional model**	**Deletion of exon 4** **(N-terminus, any functional domain)**	Cd4-Cre;PIK3C3^fl/fl^ CD4 and CD8 cells	Viable mice T cell lymphopenia, reduced T cell count Impaired autophagy in T cells	[112]

### 6.4. Class III PI3K and Its Role in Cancer

Most of the research on the role of Vps34 in cancer is restricted to studies in cell lines. In cancer cells, this enzyme promotes cell survival and proliferation by inducing autophagy. In human breast cancer cells, Vps34 activates the transcription and the activating phosphorylation of the autophagosome cargo protein p62, contributing to cell oncogenicity [113]. Vps34 inhibitors also improve the sensitivity of breast cancer cells to tyrosine kinase inhibitors sunitinib and erlotinib [114]. In renal tumor cells, the pharmacological inhibition of Vps34 with SAR405 can synergize with everolimus, an mTOR inhibitor that promotes autophagy induction, to reduce cell proliferation [115]. Vps34 regulates iron metabolism (possibly inhibiting ferroptosis and promoting autophagy-induced cell protection) which can modulate RKO colon cancer cell line proliferation/survival [116]. Vps34 contributes to EGFR translocation to the nucleus, inhibiting the pro-apoptotic activity of the Arf promoter in lung tumor cells [117].

In vivo, the genetic or pharmacological inhibition of Vps34 in mice reprograms melanoma and colorectal immune-deprived cold tumors into inflammatory tumors that secrete, amongst others, CCL5 and CXCL10 chemokines [118,119]. This process could be dependent on autophagy regulation. These tumors are then highly infiltrated with natural killer cells and CD8+ T lymphocytes which increase the efficacy of anti-PD-1/PD-L1 immunotherapies [118,119]. Thus, Vps34 activates a tumor-intrinsic immune escape process during tumor development [118,120].

The opposite functions (protection) of autophagy in cancer initiation have also been described [121], yet a clear demonstration of this protective role remains to be provided. Events hinting at such a protective role have been described: Beclin-1 overexpression in breast cancer cell line MCF-7 decreases tumor formation in nude mice, while the mutation in the domain of Beclin-1 interacting with the Vps34 complex reverses these tumor suppression functions [122]. In addition, Bif1 can activate autophagy and act as a tumor suppressor protein by interacting with Beclin-1 and UVRAG [123].

Further in vivo data are required to define the exact functions of tumor-intrinsic and -extrinsic Vps34 on tumor initiation and progression in tissue-specific models. The current published results emphasize the need to follow a cautious strategy when targeting the class III PI3K in cancer, as there are potential severe toxic effects on cardiac and hepatic functions (Table 2). Nevertheless, the partial inactivation of Vps34 by pharmacological means seems to yield beneficial actions [104].

After years of neglect, the functions of class II and III PI3Ks in physiology are now being unraveled. The analysis of genetically engineered mouse models mimicking their pharmacological inhibition and their validation in human samples could uncover the potential toxic action of their respective inhibitors (Table 1 and Table 2).

## 7. Future Directions

### 7.1. Understanding Isoform Specificity in Cancer

Although the understanding of PI3K roles and their molecular mechanisms has vastly progressed since their discovery 30 years ago, much remains to be learned. There are several gaps in our knowledge regarding the interaction and organismal roles of PI3Ks. Emerging evidence suggests that lipid pools produced at different cellular locations might not have the same physiological functions. Likewise, there is growing evidence that timing and tissue context greatly impact the type of response that a certain stimuli might elicit [124].

The PI3K isoform specificity is still not fully understood but, so far, it is attributed to different PI3K expression levels, localizations and mutations. Another hypothesis sustains that each PI3K isoform could also act on specific PIP2 sub-species. Indeed, mutations on TP53, the gene encoding for the tumor suppressor protein p53, are responsible for the modification of the PIP3 (produced by PI3Ks) acylation state [125,126]; this could affect the phospholipid membrane localization [127] and could be responsible for differential phosphorylation by each PI3K, explaining the differential effect of isoforms [17,128]. Indeed, isoform selective inhibitors differently modify the distribution of PIP3 species.

Given the key importance of tumor-cell intrinsic and extrinsic metabolism for controlling cancer progression, one key hypothesis that we are pursuing is that the selective role of class I PI3Ks is explained by their selective roles in controlling cell metabolism. This is based on the isoform-selective physiological functions of PI3K identified in metabolic organs [3] but also on the identified selective molecular effectors or cell processes controlled by each isoform [8,66].

### 7.2. Cooperation between Class I PI3K Isoforms

Cooperation and compensations between class I PI3K isoforms have been widely described; for example, compensations between PI3Kα and PI3Kβ, a redundancy between PI3Kα and PI3Kδ through receptor tyrosine kinase (RTK) or between PI3Kγ and PI3Kβ downstream G-protein coupled receptor (GPCR) [3,4] (Figure 3). Hence, we and others have described: 1—immediate compensations/redundancy between isoforms that are activated through similar mechanisms (e.g., PI3Kα and PI3Kδ downstream RTK [129], or PI3Kγ and PI3Kβ downstream GPCR [43]), or 2—delayed (in a time of 1–2 weeks) compensation/redundancy between the two ubiquitously expressed PI3Ks, PI3Kα and PI3Kβ (e.g., an overexpression of RTK, a mutation of PTEN) [130,131,132]. We recently identified a possible new mode of resistance based on a rapid rewiring network. As described above, recent data show compensatory signals via the AKT pathway upon treatment with PI3Kα or PI3Kγ-specific inhibitors in pancreatic cancer cell lines. The combined inhibition of these two isoforms in subcutaneous mice models of pancreatic cancers showed a synergistic anti-tumor effect compared to each treatment alone [8].

The cooperation between isoforms has implications in the clinical application of PI3K inhibitors.

### 7.3. Cooperation between Class I at Plasma Membrane and Class II and III at Intracellular Membranes

The different PI3K classes are mainly studied as an individual class of enzymes regulating the phosphorylation of distinct phospholipids. However, the production of those lipids is interconnected, the production of each lipid potentially impacting the others. Therefore, altering one lipid kinase might trigger knock-on effects for the other classes of PI3Ks. This cross-regulation of PI3K classes may be considered to prevent PI3K inhibitor resistance. To our knowledge, this has not been analyzed. In any case, the indirect interactions between pathways and processes controlled by distinct PI3K isoforms is recognized as a line of research that goes beyond the mere regulation of mTORC1 activity.

Undeniably, the most described interconnection between those pathways is indirect and involves the regulation of the mammalian target of rapamycin complex 1 (mTORC1). PIP3 generated by a class I PI3K allows the recruitment of phosphoinositide-dependent kinase-1 (PDK1), mammalian target of rapamycin complex 2 (mTORC2) and AKT to the plasma membrane. PDK1 phosphorylates and activates AKT which subsequently enhances mTORC1 activity. AKT and the downstream effector mTORC1 are already known to inhibit autophagy, for example by phosphorylating and reducing the expression of UVRAG, a protein contained in the Vps34 complex II. Indeed, the pharmacological inhibition of Vps34 by SAR405 reduces autophagy induced by mTOR inhibition [104]. The SUMOylation of PDK1 enhances AKT/mTORC1 activation, which inhibits the autophagic flux. However, Vps34 was known to inhibit PDK1 SUMOylation, and non-SUMOylated PDK1 directly positively regulates an autophagosome’s formation, tethering LC3 to the endoplasmic reticulum to initiate autophagy [133]. The interconnection between class I and class III-regulated mTORC1 activation was also identified upon the long-term inhibition of class I PI3K, demonstrating their interdependence. The prolonged treatment of breast cancer cells with class I PI3K or AKT inhibitors leads to the increased expression and activation of a kinase termed SGK3 which is related to AKT. Under these conditions, SGK3 is controlled by Vps34 that generates PI_(3)_P, which binds to the PX domain of SGK3 promoting phosphorylation and activation by its upstream PDK1 activator; SGK3 next substitutes AKT by phosphorylating TSC2 to activate mTORC1 [134].

Data also described indirect links between class II PI3K and pathways regulated by class I PI3K with PIK3-C2α activating mTORC1 via Raptor in adipocytes [135] and PI3K-C2β repressing mTORC1 activity via PI_(3,4)_P2 production [136]. Interestingly, the combined pharmacological inhibition of PI3Kβ with PI3K-C2α downregulation significantly reduced PC3 cell migration, better than with each treatment alone [135]. This suggests that these two isoforms could regulate cell proliferation by distinct molecular pathways.

Previous data showed that PI3Kβ and Vps34 can interact with each other directly or indirectly via Rab5 [137]. On the other hand, PTEN, which preferentially inhibits PI3Kβ products, can be localized in intracellular vesicles and become inactive via its binding to PI_(3)_P whose level is regulated by Vps34 [138]. Hence, PI3Kβ inhibitor efficiency might be linked to Vps34 basal levels of activity (Figure 3).

In spite of the scarce knowledge, emerging data indicate that class I, II and class III-controlled pathways tightly regulate each other and can act as a rheostat for their respective pathways and functions (i.e., autophagy and AKT/mTOR pathway, but maybe other processes such as macropinocytosis of MHC-I surface receptors) (Figure 3). Although, it is important to note that those cross-regulations were demonstrated in different cell types and contexts. Limited information is available in cancer cells and in in vivo preclinical cancer models. If those regulations are confirmed in cancer settings, further combination treatments could be designed such as the targeting of Vps34 and class I PI3K to control tumoral immunity. Indeed, Vps34 inhibition triggers PD-L1 expression [118] while PI3Kδ induces the expression of PD-1 in CD8+ cells [139]. The combination targeting of class I and II or class I and class III PI3Ks has never been achieved in preclinical models. Further in vitro and in vivo studies are required to determine how different PI3K classes interact in the context of an organ or a tumor. These data are crucial for better targeting of isoforms’ specific functions (Figure 3).

**Figure 3 cancers-15-00784-f003:**
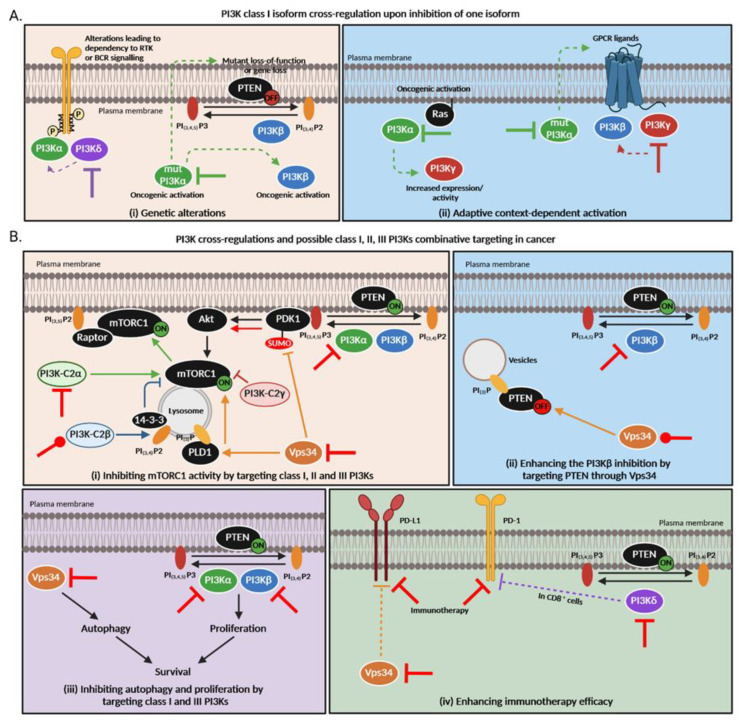
PI3K cross-regulations and possible class I, II and III PI3Ks combination targeting in cancer. (**A**). PI3K class I isoform cross-regulations upon inhibition of one isoform. Thick lines = primary inhibition, dashed arrows = subsequent regulations. (**B**). PI3K cross-regulations among different classes are still poorly understood but based on this knowledge, new therapeutic strategies arise to overcome treatment resistance in cancer. (**i**) All PI3K classes directly or indirectly regulate mTORC1 activity. Class I PI3Kα, β, γ and δ phosphorylate PIP2 into PIP3 which allow the recruitment of PDK1 to the plasma membrane. PDK1 then activates AKT which subsequently enhances mTORC1 activity. The latter can inhibit PI3K in a negative feedback. Class II PI3K-C2α activates mTORC1 and favors its translocation to the plasma membrane via Raptor association to PIP2 [135,140]. On the contrary, PI3K-C2β produces PI_(3,4)_P2, allowing the link of 14-3-3 which represses mTORC1 [136]. Loss of PI3K-C2γ is responsible for an mTOR activation [96]. Vps34 allows the amino acid activation of the phospholipase D1 (PLD1) which then translocates to the lysosome and stimulates mTORC1 activity [141] but can also inhibit PDK1 sumoylation which results in less AKT and mTOR activation. (**ii**) Vps34 regulates autophagy while class I PI3Ks modulate proliferation. Both mechanisms are important for cancer cell survival. (**iii**) Class III PI3K Vps34 can indirectly modulate PI3Kβ activity by allowing PTEN relocalization to intracellular PI_(3)_P vesicles whose level is regulated by Vps34 [138]. (**iv**) Vps34 inhibition triggers PD-L1 expression [118] while PI3Kδ induces the expression of PD-1 in CD8+ cells [139]. “ON” means that the protein is activated while “OFF” means it is inactivated or absent. Red inhibition or activation lines correspond to possible combination therapies that could be inhibitory or activatory. Of note, all these pathways were demonstrated in different cell types and contexts; the large significance of these data is currently unknown.

### 7.4. Multi-Isoform Targeting in Cancer

To date, multi-isoform targeting tools are under development but are limited to the targeting of several class I isoforms. Taselisib, a β-sparing inhibitor (targeting all class I isoforms but PI3Kβ) has been recently tested in a phase I clinical trial in *PIK3CA*-mutated cancers [12]. Only proof-of-concept for the simultaneous targeting of PI3Kδ and Vps34 in AML, CLL and Burkitt lymphoma cell lines was demonstrated [142].

A solution to increase the possibilities of combinations could be to design combi/hybrid molecules, a combination of several existing drugs attached with a linker and designed to target several proteins with less adverse effects and a controlled pharmacokinetic [143,144,145] (Figure 4). Combi/hybrid molecules have been developed to target both PI3Ks and histone deacetylase (HDAC) such as CUDC-907 with relevant anti-tumor activity in neuroblastoma or AML [146,147]. However, no combi/hybrid molecule has yet been developed to target simultaneously several PI3K isoforms among different classes. Those molecules harbor several advantages, such as the possibility to choose linkers whose cleavage can be specifically induced by the intrinsic tumor metabolism, possibly reducing toxicity. Compared to normal cells, tumor cells overactivate class I PI3Ks. Therefore, combi/hybrid molecules targeting class I PI3K and class II or III could also act as tumor cell hunters with higher selectivity and lower toxicities. We would then be able to exclusively target the relevant isoforms and improve the efficacy of PI3K therapies in cancer by using novel methods to determine the potency of combination targeted therapies.

## 8. Conclusions

The entire family of PI3Ks regulate key mechanisms in cancer. The pathways controlled by class I PI3Ks are altered in more than 50% of all cancers (solid and hematopoietic). Notwithstanding the fact that the specific roles of class I PI3K isoforms become better characterized in each cancer subtype [3], more research is needed to clearly decipher the functions of class II and class III PI3Ks in cancer, as there is emerging evidence that they play a crucial role in tumor biology.

The strategy consisting of targeting all class I PI3Ks with pan-PI3K inhibitors rapidly showed limitations due to toxicities in combination treatments and an increased resistance to treatment [8], emphasizing the need to specifically target the key PI3K isoforms in each cancer subtype [4]. Currently, PI3Kα-specific inhibitors are administered to *PIK3CA*-mutated tumors in combination with hormonotherapy; the clinical use of PI3Kα inhibitors could be extended to cancer with global PI3K/AKT activation (e.g., pancreatic cancer due to *KRAS* mutation) but not in single therapy and possibly with PI3Kγ inhibitors. PI3Kβ-specific inhibitors should not be restricted to *PTEN*-deficient cancers, but the key predictive markers of their use and the treatment that needs to be combined are not clearly defined. 

Regarding the targeting of PI3Kδ, which is efficient to target hematopoietic cells but also as an immunotherapy, the main challenge is to handle the toxicity. Approaches such as intermittent doses [65], but also a better determination of the efficacy, need to be reviewed [148]. Analyzing the toxicity of the compounds and their metabolites when used in combination is necessary to lead those pharmaceutical compounds to successful clinical trials (Figure 5).

Finally, given the emerging evidence that class I and II/III are cross-regulating their downstream cellular processes and cooperating to promote cancer cell proliferation, survival and migration, one underexplored strategy could be to specifically target class I, II and/or III PI3K isoforms through combination therapies (Figure 5). However, those combinations may increase the adverse effects in patients, possibly leading to the arrest of the treatment. While emerging evidence was found in pancreatic cancer [96], a basic knowledge of the roles of class II and III PI3Ks in tumor biology is still missing to enable those strategies.

In parallel to the clinical assessment of class I PI3K inhibitors, future basic and preclinical research should provide the proof-of-concept for developing combination treatment strategies targeting several PI3K isoforms amongst the three classes.

## Figures and Tables

**Figure 1 cancers-15-00784-f001:**
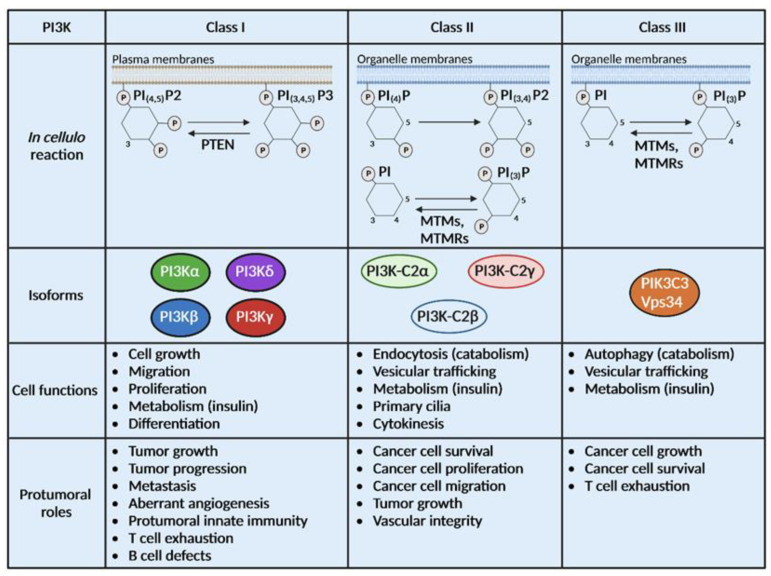
PI3Ks are divided into 3 classes with different substrates and functions. Class I, II and III PI3Ks catalyze the same biochemical reaction on distinct substrates localized at the plasma membrane or the organelle membrane. Some phosphatases can catalyze the opposite reaction of PI3Ks such as Phosphatase and Tensin homolog (PTEN), myotubularin (MTM) and MTM-related protein (MTMR). Class I is composed of 4 isoforms (PI3Kα, β, δ and γ), class II of 3 isoforms (PI3K-C2α, β and γ) and class III has only one isoform, Vps34. Physiological and pro-tumoral functions of class I PI3Ks are well described while limited information is available on the roles of class II and III PI3Ks in cancer.

**Figure 2 cancers-15-00784-f002:**
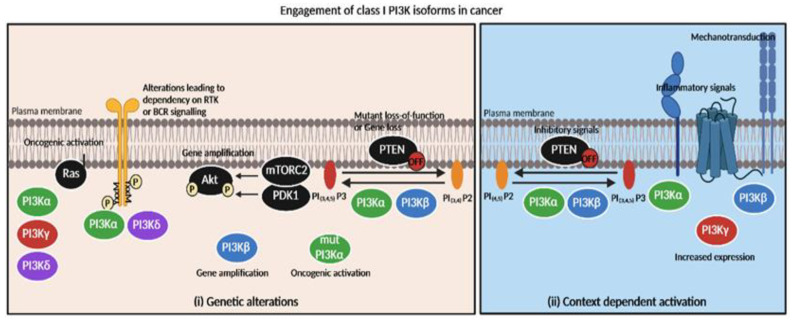
Engagement of class I PI3K isoforms in cancer depending on genetic alterations or context-dependent activation. Tumors’ specific dependency on PI3K inhibitors is mainly based on (**i**) genetic alterations and (**ii**) context-dependent activation of isoforms. (**i**) Oncogenic activation of PI3Kα catalytic domain leads to PI3Kα sensitivity while PI3Kβ gene amplification or PTEN loss is responsible for PI3Kβ inhibitors’ efficacy. RAS oncogenic activation directly couples with PI3Kα,γ or δ isoforms (but not PI3Kβ) [49,50]. (**ii**) GSK3β-dependent inhibitory phosphorylation of PTEN could possibly increase sensitivity to PI3Kα or PI3Kβ inhibitors. Inflammatory context (e.g., TNFα [17], GPCR ligand S1P [43]) activates PI3Kα or PI3Kβ/PI3Kγ, respectively. Mechanical signals possibly promote PI3Kβ activation [40]. Increased expression of PI3Kγ promotes PI3K activation [51].

**Figure 4 cancers-15-00784-f004:**
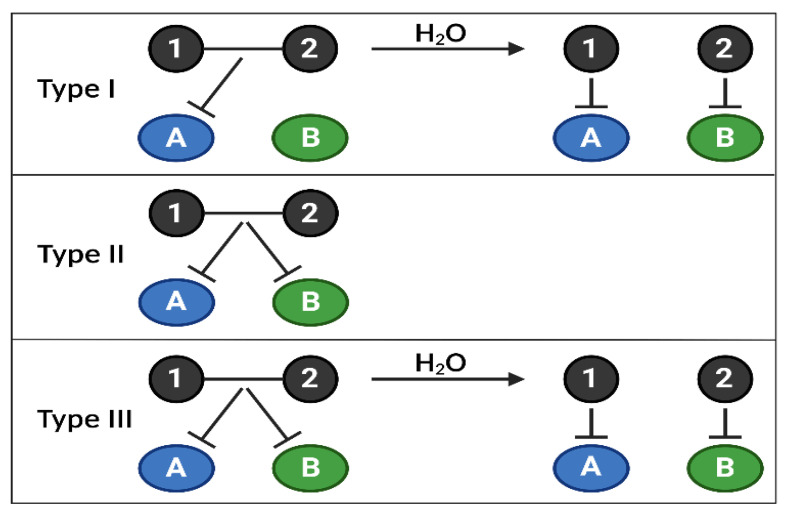
Combi/hybrid molecules, annihilating several targets at once. The use of targeted therapies in clinics has highlighted the appearance of frequent treatment resistance due to the activation of compensatory feedbacks, requiring inhibiting several signaling nodes at once. Combi- or hybrid molecules have been developed to overcome these limitations by targeting several biological targets (kinases, DNA, etc.) with one entity, allowing better control of the pharmacokinetics and pharmacodynamics of drug combinations. Currently, there are 3 types of combi/hybrid molecules, depending on their mechanism of action. Type I can block target A as an intact molecule but is required to be hydrolyzed to block A and B; Type II can block targets A and B without being hydrolyzed; Type III can block targets A and B as an intact molecule or after being hydrolyzed. Combi/hybrid molecules are one of the proposed novel therapeutic solutions to target specific PI3K isoforms among the 3 classes, depending on their relative importance in a type of cancer, using only one molecule.

**Figure 5 cancers-15-00784-f005:**
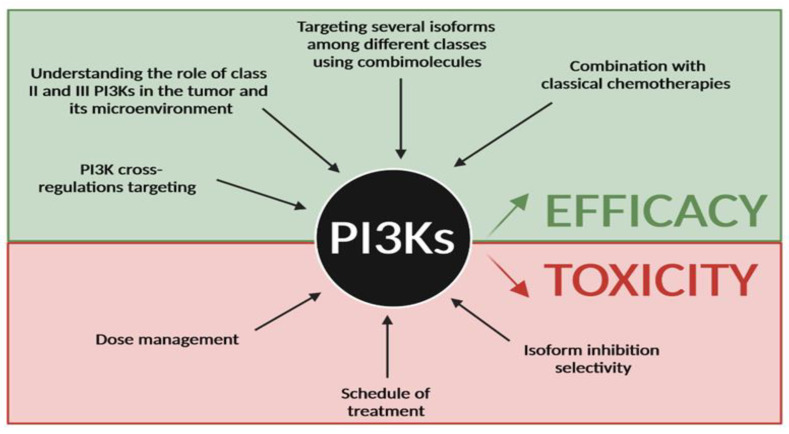
Future direction for PI3K targeting in cancer. Efforts have to be made to improve efficacy of PI3K inhibitors and reduce their toxicity in cancer treatment. Improving efficacy will require better knowledge of PI3K cross-regulations and the role of class II and III in cancer, to target several PI3K isoforms among the 3 classes and to combine existing PI3K inhibitors with classical chemotherapies. Reducing toxicity will go through a better dose management, different schedules of treatment and a better understanding of isoform specificity in organs and in cancer subtypes.

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
