# Peer review of "Targeting Class I-II-III PI3Ks in Cancer Therapy: Recent Advances in Tumor Biology and Preclinical Research"

_cancers, 2023, doi:10.3390/cancers15030784_

Round 1

Reviewer 1 Report (Previous Reviewer 2)

* The manuscript improved significantly and it will be accepted after moderate English revision by a native English speaker.

Author Response

Please find a revised manuscript read by a native English speaker.

Many thanks for your feedback.

Best wishes,

Julie Guillermet-Guibert, PhD

This manuscript is a resubmission of an earlier submission. The following is a list of the peer review reports and author responses from that submission.

Round 1

Reviewer 1 Report

The review by Thibault et al. describes the advances in the roles of the different classes of phosphoinositide-3-kinases in cancer.

Overall, this contribution gives only a partial and incomplete picture of the field with a little critical contribution. Furthermore, this very interesting topic is analyzed in a very superficial and sketchy way. There is a lot of emphasis on animal models but scarce clinical and human data.

Figure 1 is unclear and misleading. There is no evidence that class II produces PIP2 (should be PI3,4P2), at least directly, since in vitro the main substrate is PI and not PI4P. What is the meaning of the two arrows with anabolism and catabolism?

There is no evidence indicating that each PI3K isoform could also act on specific PIP3 sub-species. It is true that the mutational status of TP53 may affect the length of the fatty acid chain of PIs but the authors fail to give a satisfactory description of the concepts. In addition, key references are missing throughout the manuscript and very often reviews are cited and not the original manuscript. The reference(s) reporting the effect of TP53 status on PIP3 acylation state is missing. For instance, the authors should check carefully the recent contribution authored by Hawkins P.

The view that PI3Kgamma has only a role in hematologic cancers is outdated and critical references on the role of this isoform in a solid tumour, such as pancreatic cancer, are missing.

As mentioned above a bit more description of the substrate preference of class II PI3K is required. The potential role of class II PI3K in cancer is largely overlooked with key references omitted. Annoyingly, the main citation here is that of a review. The author should note that conclusions made primarily on animal models and not on clinical data are largely misleading. For instance, reference 55 conclusions are not supported by the dataset showing that class II PI3K gamma is overexpressed in pancreatic cancer and survival is inversely correlated with high protein expression which is exactly the inverse conclusion made in that contribution.

There are many vague and enigmatic sentences such as those at the end of page 7.

The Conclusion section does not convey a take-home message and future perspectives are missing. More importantly, the critical opinion of the authors is clearly missing. The meaning of Figures 3 and 4 and the pertinence in this context of the combination therapy is obscure.

Author Response

Reviewer 1:

The review by Thibault et al. describes the advances in the roles of the different classes of phosphoinositide-3-kinases in cancer.

Overall, this contribution gives only a partial and incomplete picture of the field with a little critical contribution. Furthermore, this very interesting topic is analyzed in a very superficial and sketchy way.

We thank the reviewer for acknowledge the importance of the topic. We agree with the reviewer that the review is not a comprehensive analysis of the all the articles on class I, II and III. The choice of the literature review was the following: for the class I, we focused on the publications made in the last 5 years; older articles are cited by the mean of reviews. For the class II and III, we agree that analyzing original articles of the last 5 years will not give sufficient literature for an interesting discussion. In this novel version of the manuscript, we added the seminal articles that are more ancient (>5 years).

We agree that our critical contribution was not properly highlighted; it was scattered throughout the manuscript. We have now clearly indicated when we provide our opinion on the field – see also novel paragraph “futures directions” at the end of the review.

There is a lot of emphasis on animal models but scarce clinical and human data.

This was modified accordingly. We agree that it is important to give the clinical context in which this basic and preclinical research is performed.

Figure 1 is unclear and misleading. There is no evidence that class II produces PIP2 (should be PI3,4P2), at least directly, since in vitro the main substrate is PI and not PI4P. What is the meaning of the two arrows with anabolism and catabolism?

The intent of the figure was improved. In the review, we focus on the proved lipid products in cellulo. PI3,4P2 is a form of PIP2 that was analyzed in cellular vesicles and produced by class II PI3Ks (PMID: 26888746). We agree that in vitro class I, II and III phosphorylate PI, class I and II PI-4-P and class III PI-4,5-P2 (PMID: 19644473). In our opinion, there is strong evidence that class II PI3K produces a form of PIP2 (PI-3,4-P2) in cellulo (PMID: 35809565).

There is no evidence indicating that each PI3K isoform could also act on specific PIP3 sub-species. It is true that the mutational status of TP53 may affect the length of the fatty acid chain of PIs but the authors fail to give a satisfactory description of the concepts. In addition, key references are missing throughout the manuscript and very often reviews are cited and not the original manuscript. The reference(s) reporting the effect of TP53 status on PIP3 acylation state is missing. For instance, the authors should check carefully the recent contribution authored by Hawkins P.

Many thanks for this comment; this was modified accordingly.

The view that PI3Kgamma has only a role in hematologic cancers is outdated and critical references on the role of this isoform in a solid tumour, such as pancreatic cancer, are missing.

We agree with the reviewer and have added a specific paragraph on this topic.

As mentioned above a bit more description of the substrate preference of class II PI3K is required. The potential role of class II PI3K in cancer is largely overlooked with key references omitted. Annoyingly, the main citation here is that of a review. The author should note that conclusions made primarily on animal models and not on clinical data are largely misleading. For instance, reference 55 conclusions are not supported by the dataset showing that class II PI3K gamma is overexpressed in pancreatic cancer and survival is inversely correlated with high protein expression which is exactly the inverse conclusion made in that contribution.

We have detailed the substrate preference of class II PI3Ks. We now do not refer only to the review anymore.

We agree that while PI3KC2α is essential in mice, human patients with inherited homozygous null mutations in the PIK3C2A gene display congenital syndromic features, including kidney failure and cataract, due to early senescence of cells in the associated tissues (Gulluni, F. et al. PI(3,4)P2-mediated cytokinetic abscission prevents early senescence and cataract formation. Science 374, eabk0410 (2021); Tiosano, D. et al. Mutations in PIK3C2A cause syndromic short stature, skeletal abnormalities, and cataracts associated with ciliary dysfunction.PLoS Genet. 15, e1008088 (2019)).

With regards to the expression of PIK3C2G in exocrine pancreas. Braccini et al, (PMID: 26100075) used a murine mouse model where LacZ expression was under control of PIK3CG promoter at the locus of the gene (see in word file attached the Suppl. Data of the article).

With regards to the match between mouse model and human patient data in cancer context, De Santis MC et al (ex-reference 55, now reference 96) show clear genetic proof that decreased expression of PIK3C2G accelerates tumor progression induced by mutant Kras and mutant p53 in the pancreas, and that there is a heterogeneity of expression of PIK3C2G in Human samples (See word file attached).

There are many vague and enigmatic sentences such as those at the end of page 7.

This was modified accordingly.

The Conclusion section does not convey a take-home message and future perspectives are missing. More importantly, the critical opinion of the authors is clearly missing. The meaning of Figures 3 and 4 and the pertinence in this context of the combination therapy is obscure.

We have improved those paragraphs accordingly, to enlarge the concept conveyed by Ex-Figure 3 (New Figure 5) to all the pharmacokinetics and pharmacodynamics issues related to PI3K inhibitor clinical trials. We have changed the position of Ex-Figure 4-new-Figure 4 in the paragraph “Future direction” to better explain the pertinence of these paragraphs.

Reviewer 2 Report

The review is not well organized, with too short and too-long paragraphs and low-provided references. Please, reorganize well and submit again.

Author Response

Reviewer 2:

The review is not well organized, with too short and too-long paragraphs and low-provided references. Please, reorganize well and submit again.

Please find our novel version of our manuscript completely re-organized (see tracking version).